# Rapid Purification and Formulation of Radiopharmaceuticals via Thin-Layer Chromatography

**DOI:** 10.3390/molecules27238178

**Published:** 2022-11-24

**Authors:** Travis S. Laferriere-Holloway, Alejandra Rios, Giuseppe Carlucci, R. Michael van Dam

**Affiliations:** 1Department of Molecular & Medical Pharmacology, David Geffen School of Medicine, University of California, Los Angeles, CA 90095, USA; 2Crump Institute for Molecular Imaging, University of California, Los Angeles, CA 90095, USA; 3Physics and Biology in Medicine Interdepartmental Graduate Program, David Geffen School of Medicine, University of California, Los Angeles, CA 90095, USA; 4Ahmanson Translational Theranostics Division, Department of Molecular & Medical Pharmacology, University of California, Los Angeles, CA 90095, USA

**Keywords:** radiopharmaceuticals, microscale radiosynthesis, thin-layer chromatography, miniaturization, preparative TLC

## Abstract

Before formulating radiopharmaceuticals for injection, it is necessary to remove various impurities via purification. Conventional synthesis methods involve relatively large quantities of reagents, requiring high-resolution and high-capacity chromatographic methods (e.g., semi-preparative radio-HPLC) to ensure adequate purity of the radiopharmaceutical. Due to the use of organic solvents during purification, additional processing is needed to reformulate the radiopharmaceutical into an injectable buffer. Recent developments in microscale radiosynthesis have made it possible to synthesize radiopharmaceuticals with vastly reduced reagent masses, minimizing impurities. This enables purification with lower-capacity methods, such as analytical HPLC, with a reduction of purification time and volume (that shortens downstream re-formulation). Still, the need for a bulky and expensive HPLC system undermines many of the advantages of microfluidics. This study demonstrates the feasibility of using radio-TLC for the purification of radiopharmaceuticals. This technique combines high-performance (high-resolution, high-speed separation) with the advantages of a compact and low-cost setup. A further advantage is that no downstream re-formulation step is needed. Production and purification of clinical scale batches of [^18^F]PBR-06 and [^18^F]Fallypride are demonstrated with high yield, purity, and specific activity. Automating this radio-TLC method could provide an attractive solution for the purification step in microscale radiochemistry systems.

## 1. Introduction

In the last decade, positron-emission tomography (PET) has led to many advances in disease characterization [1,2], drug development [3,4,5], and monitoring treatment efficacy for various diseases [6,7]. While numerous short-lived radionuclides may be used to label biologically active radiotracers, fluorine-18 remains by far the most common due to its high positron decay ratio (97%), short half-life (109.8 min), low positron energy (635 keV), and wide availability [8,9,10].

The production of ^18^F-labelled radiopharmaceuticals typically involves a late-stage radiofluorination method involving the reaction between [^18^F]fluoride or a prosthetic group labelled with F-18, and a precursor, followed in some instances by the deprotection of functional groups. Subsequently, purification of the crude radiopharmaceutical is required to ensure that all unreacted precursors, reaction by-products, solvents, and other reagents (e.g., phase-transfer catalysts), are removed. The high structural similarity of precursors and byproducts to radiopharmaceuticals, along with the vast precursor excess typically used in radiosyntheses to ensure efficient reaction kinetics, impose significant challenges for the purification process.

Several chromatographic approaches are currently used to purify radiopharmaceuticals, including solid phase extraction (SPE) and high-performance liquid chromatography (HPLC). While these chromatographic methods are both versatile and compatible with a variety of stationary phases (e.g., reverse-phase C18 [11,12], size exclusion (SE) [13,14], and ion exchange (IEX) [15]), they differ in complexity and performance. SPE is generally rapid, but separation resolution is generally regarded as low [16]. A further downside is that developing a suitable SPE-based purification protocol can take considerable time and effort. So far, SPE has only successfully been used to purify a handful of ^18^F-labeled radiopharmaceuticals [17,18,19,20,21]. HPLC has high resolution and is used to purify the vast majority of radiotracers. Still, it is time-consuming, bulky, expensive, and often requires a downstream re-formulation process due to bio-incompatible mobile phases [22,23]. In some instances, HPLC can be performed with bio-compatible (e.g., ethanolic) mobile phases, although increased backpressure can become an issue. Another approach that has been used to purify radiopharmaceuticals is molecular imprinting chromatography [24]. However, this technique requires a unique stationary phase for each radiopharmaceutical and is not widely used.

Recently, our group and others have shown that microscale synthesis methods enable efficient reactions while enabling a vast reduction of reagent masses [25,26,27,28,29]. Consequently, the quantity of impurities is drastically reduced, and it appears in some cases that the number of different impurities may also be reduced [28,30]. These factors may allow lower-resolution forms of purification to be employed in the purification of microscale-produced radiopharmaceuticals. For example, purification has been performed using microscale SPE for [^18^F]FDG [31,32,33] and [^18^F]FLT [34,35]. It has also been attempted for microfluidically-produced [^18^F]Fallypride [36], but sufficient chemical purity was not achieved, suggesting that microscale and conventional SPE may have similar limitations of versatility due to the low separation resolution. Alternatively, our group and others have shown that conventional semi-prep HPLC columns can be replaced with analytical scale columns [37,38], enabling faster purification, higher resolution, and reduced volume of collected pure fraction (enabling faster downstream formulation). However, the continued need for a bulky and expensive instrument to perform purification undermines many of the advantages of microfluidic radiosynthesis.

To overcome these challenges, we propose using thin-layer chromatography (TLC) as a more compact, rapid, and lower-cost way to purify microfluidically-produced radiopharmaceuticals. Purification via TLC is not new and is often used in the pharmaceutical industry for the crude synthesis of candidate molecules [39]. Utilizing preparative TLC plates, crude products are separated, then the product-binding sorbent is removed from the plate and extracted in organic solutions for subsequent processing. Though separations in the pharmaceutical industry usually involve long TLC plates and lengthy separation times, which are incompatible with the production of short-lived radiopharmaceuticals and the goals of miniaturizing the entire radiosynthesis processes, the masses involved in batches of radiopharmaceuticals are far smaller. We hypothesized that short, analytical-scale plates might be suitable for purifying radiopharmaceuticals (Figure 1).

Radio-TLC is already widely used in radiochemistry to analyze small samples (e.g., 1 µL) of radiopharmaceuticals. By making use of high-resolution imaging-based readout (e.g., Cerenkov luminescence imaging; CLI and UV imaging), our group has recently shown that separation resolution comparable to radio-HPLC can be achieved on analytical scale TLC plates with very short separation distances (4 cm) and short separation times (<4 min) [40]. In addition to being rapid and having high resolution, TLC is very versatile. We recently adopted the PRISMA algorithm [41] for efficiently optimizing mobile phase compositions to achieve high separation of a wide variety of radiopharmaceuticals from their radioactive and non-radioactive impurities [42]. This paper shows the feasibility of using analytical-scale TLC as a compact, rapid, and high-resolution method for the purification of microfluidically-produced (i.e., low mass scale, low volume) radiopharmaceuticals.

## 2. Results

### 2.1. Performance of TLC at the Scale of Crude Reaction Mixtures

When performing TLC analysis of radiopharmaceuticals, typically, only a small sample volume (0.5 or 1.0 μL) is spotted on the plate via a capillary or pipette. In contrast, the volume of the collected crude product from microscale reactions is on the order of 40–60 μL [43], all of which need to be loaded onto the TLC plate to use this as a purification method.

We have previously used the PRISMA algorithm to establish suitable TLC mobile phases for baseline separation of [^18^F]PBR-06 from radioactive and non-radioactive impurities in crude reaction mixtures (1 µL sample) and for [^18^F]Fallypride from its impurities (1 µL sample) [44]. While the separation resolution is expected to suffer by increasing the sample volume and mass, the degree of resolution reduction needs to be quantified.

To study the effect of sample mass without significantly changing the size of the sample spot, we loaded samples by pipetting crude [^18^F]PBR-06 in 1 µL increments onto the origin while heating the TLC plate with a heat gun (90 °C setting), allowing each droplet to dry (~2 s) before adding the next. Comparison of samples before and after heating showed no changes, including no signs of decomposition. For [^18^F]PBR-06, we discovered that the chromatographic resolution of [^18^F]PBR-06 from its nearest impurity decreased from 2.2 to 1.0 when increasing the total volume of the spotted crude product from 6 to 60 μL. Notably, 60 μL corresponds to an entire batch of crude radiopharmaceutical. When performed manually, sample deposition with this approach took ~5 min to load a 60 µL sample.

We next tried applying the 60 μL volume in a streak rather than a single spot. Each 20 μL portion was deposited as a ~20 mm long line along the origin and then dried at 90 °C (~5 s) before applying the next streak in the same location. By spreading out the mass amount of product over a greater width of the separation medium, the chromatographic load is decreased, achieving nearly the same resolution (2.0) as the plate spotted with only 6 µL. The 60 µL sample could be deposited in <2 min by streaking.

We also tried spotting a 60 µL sample to TLC plates containing a concentrating zone. A large volume can be deposited as a single spot within the concentrating zone. During development, it will be concentrated into a thin line at the boundary of the concentrating zone before its migration and separation within the separation zone. Using this method, the resolution was 1.7. While we expected the resolution to be similar to the streaking approach, the observed resolution may be slightly lower because the plate was a HPTLC plate, which has a thinner sorbent layer (150 μm) than the analytical plates used for other samples (250 μm).

These results are summarized in Figure 2 and Table 1. Due to the high performance of the sample streaking method, in conjunction with analytical TLC plates, they were used for the remainder of the study.

In addition to evaluating separation resolution, losses during the sample application process were evaluated. Measurements using a calibrated ion chamber (CRC 25-PET, Capintec, Florham Park, NJ, USA) revealed a ~10–20% loss of initial sample activity on the pipette tip and Eppendorf tube originally containing the crude radiopharmaceutical. The loss could be reduced to <1% if the Eppendorf and pipette tip were rinsed with 20 μL of 9:1 MeOH:H_2_O (*v*/*v*). When applied during the sample streaking method, the additional rinse volume did not affect the chromatographic resolution.

### 2.2. Efficiency of Radiopharmaceutical Collection from the TLC Plate

After separation, the product must be collected from the plate. We elected to use a process of scraping the silica stationary phase from the plate in the region of the desired band, followed by extraction of the product into a buffer.

The effectiveness of collecting the sorbent-bound radiopharmaceutical for [^18^F]PBR-06 and [^18^F]Fallypride is demonstrated in Figure 3. Each TLC plate was streaked with 60 µL of crude product, developed, and then measured by a dose calibrator. A CLI image of the plate was then obtained, and the relative abundance of radiochemical species was determined using region of interest (ROI) analysis, as previously described [40,45]. Combining these two measurements, we could estimate the initial quantity of radioactivity corresponding to the radiopharmaceutical product on the TLC plate. Initially, scraping of the sorbent at the position of the radiopharmaceutical band was performed via a small spatula. The sorbent (a fine powder) was collected onto weighing paper and then transferred into a SPE tube. However, using this method, >20% of the radiopharmaceutical activity (and sorbent) could be lost. Instead, we used a piece of plastic tubing with a beveled tip as the scraper. We connected the other end of the tubing through an empty SPE tube fitted with a 0.2 μm frit (Figure 1) to a vacuum source to capture the removed sorbent more efficiently. The entirety of the scraping process took <2 min to complete. The use of vacuum minimized the chance for the dispersing the radioactive powder into the air. Comparison of the collected sorbent activity of the product from the TLC plate (measured via dose calibrator) to the estimate of initial radioactivity of the radiopharmaceutical on the plate indicated that the sorbent-bound product was collected with >97% efficiency for both [^18^F]PBR-06 and [^18^F]Fallypride (Table 2, rows 1 and 3). Additional CLI images of the TLC plates were obtained after the scraping process. ROI analysis showed that the region of the plate corresponding to the product contained ~0% of the initial radioactivity (Figure 3), confirming that the silica removal process is quantitative.

### 2.3. Efficiency of Radiopharmaceutical Extraction from the Collected Sorbent

Finally, the purified radiopharmaceutical needs to be separated from the sorbent. This is accomplished by flowing liquid through the sorbent and capturing the eluted liquid while the particles remain trapped by the frit. For this step, the output of the SPE tube (containing the sorbent-bound product) is connected through a sterilizing filter (0.2 μm) to a sterile septum-capped product vial. Vacuum is applied to a sterile filter connected to the vent port of the product vial. The end of the tubing used for scraping the sorbent is then dipped into an Eppendorf tube filled with extractant solution, effectively rinsing the sorbent collection path and eluting the radiopharmaceutical from the collected sorbent.

To avoid needing a later downstream reformulation step, we evaluated the ability to extract the product from the sorbent into biocompatible solutions. Initially, saline was used to extract [^18^F]PBR-06 and [^18^F]Fallypride from the sorbent. Using 1 mL of saline, the extraction efficiency was >95% for both tracers (Table 2, rows 1 and 3). While extraction efficiency with the model radiopharmaceuticals was high, some radiopharmaceuticals require additives, such as EtOH, to improve solubility. For this reason, we also explored the use of other bio-compatible solvents for extraction. Using 100 μL of EtOH, followed by 900 μL of saline, it was possible to extract >97% of the product from the sorbent for both tracers ([^18^F]PBR-06 and [^18^F]Fallypride) (Table 2, rows 1 and 3). Flowing the additional 900 μL of saline through the sorbent provided a final formulated product with <10% EtOH (*v*/*v*).

We achieved very high overall radiochemical yield (RCY) for both radiopharmaceuticals with the combination of droplet radiosynthesis and TLC-based purification/formulation (Table 2). Compared to our prior reports of droplet radiosyntheses that used analytical-scale HPLC purification (with purification efficiency of ~80%), the efficiency of the TLC purification and formulation process was significantly higher (nearly quantitative), leading to higher overall radiochemical yield. In particular, a prior report of droplet-based [^18^F]PBR-06 production showed high crude RCY (94 ± 2%, *n* = 4), but due to losses during HPLC purification, the isolated RCY was only 76% (*n* = 1) [27], and further losses would have been expected during downstream formulation, which was not performed in that study. Similarly, a prior report of droplet-based [^18^F]Fallypride production exhibited high crude RCY (96 ± 2%, *n* = 4), but, due to losses during HPLC purification, the isolated yield was 78% (*n* = 1) [26].

Notably, the entire purification and formulation process with the TLC method was very fast and took <10 min to complete (2 min for sample spotting, >4 min for TLC plate development, 2 min for silica removal, and 2 min for radiopharmaceutical extraction and filtration).

### 2.4. Scale-Up to Clinical Quantities

The ability of the TLC method to purify radiopharmaceuticals at clinically relevant levels were explored. For droplet-based radiosynthesis, scale-up is achieved by simply increasing the amount of radioactivity in the synthesis and does not require increasing the reaction mass scale [38]. For this reason, the chromatographic resolution of the TLC method should not be impaired when utilizing greater activity scales. Indeed, scaling up the amount of radioactivity led to the efficient purification of clinically relevant activity levels of [^18^F]PBR-06 and [^18^F]Fallypride (Table 2, lines 2 and 4). Automating the TLC purification procedure may allow more activity scales to be purified.

### 2.5. Quality Control Testing of Purified [^18^F]PBR-06

A series of selected key quality control (QC) tests were performed to assess the safety and purity of the radiopharmaceuticals purified (and formulated) using the TLC method. Tests performed include appearance (color, clarity), pH, residual phase transfer catalyst, residual solvents, radiochemical purity, chemical purity, and radiochemical identity.

Radiochemical and chemical analyses were performed using HPLC (Figure 4). When we initially analyzed TLC-purified [^18^F]Fallypride (Appendix A), we noticed some impurities at early retention times in the UV channel and confirmed that these peaks came from the TLC plate itself. By pre-cleaning the TLC plates, these impurity peaks could be removed (Appendix A). When using pre-cleaned TLC plates, radiochemical and chemical purity standards suitable for injection were achievable.

The results of these and additional tests (described in the Appendix A) for three consecutive batches of TLC purified [^18^F]PBR-06 are summarized in Table 3. The results suggest that this method could potentially be used to produce tracers for clinical use.

Due to the silica sorbent’s integral role in the TLC-purification process, we were concerned that some silica could end up in the final formulation, either as small nanoparticles that pass through the frit and filter or through solubility of silica in aqueous solutions [46]. To determine levels of residual silica, we used ICP-MS to measure Si content of samples that were first digested in HNO_3_ to ensure any particulate silica was captured into the solution (see Appendix A). Si was not detected for the formulated tracer samples (limit of detection 0.83 ng/mL). While the complete elimination of silica in the final radiopharmaceutical formulation cannot be confirmed, it can be concluded that the residual amount is extremely low.

## 3. Discussion

A significant advantage of the TLC-based purification approach described is its high-speed operation. In addition to the rapid separation via TLC, for the radiopharmaceuticals tested, the purified tracer could be recovered in saline (or a mixture with <10% EtOH), eliminating the need for a downstream reformulation step. Current microscale radiopharmaceutical production protocols generally rely on HPLC purification, followed by a separation formulation step performed via solid-phase extraction on a reversed-phase cartridge or via solvent evaporation followed by resuspension in an injectable buffer, requiring 30–60 min to complete [47,48,49]. In contrast, for the TLC purification (and formulation) method, these steps were completed in <10 min for both [^18^F]PBR-06 and [^18^F]Fallypride. Based on the half-life of fluoride-18, an additional 20–50 min of overall synthesis time would lead to a 12–27% loss of product. Furthermore, during HPLC and cartridge- or evaporation-based reformulation, activity losses are typically substantially higher than the 3–5% loss observed here.

We found the product band retention factors and band heights to be remarkably consistent from run to run for both the [^18^F]PBR-06 and [^18^F]Fallypride product bands (i.e., (R_f_ = 0.66 ± 0.01, band height = 0.22 ± 0.05 cm, *n* = 7), (R_f_ = 0.91 ± 0.01, band height = 0.31 ± 0.05 cm, *n* = 4), respectively). This allowed us to mark the TLC plate in advance with the expected position of the product band, allowing the sorbent collection without imaging the TLC plate. Batches processed in this fashion had high efficiency (low loss of product) and high chemical and radiochemical purity, equivalent to batches that relied on imaging. This observation suggests that, for well-developed methods, the TLC plate imaging step can potentially be skipped, simplifying the apparatus and procedure. To use this technique reliably requires adequate separation of the desired radiopharmaceutical band from impurity bands (both radioactive and non-radioactive impurities). The mobile phases used for the separation of [^18^F]PBR-06 and [^18^F]Fallypride from impurities were optimized using a recently-reported methodology (PRISMA) to maximize the resolution between the radiopharmaceutical and nearest impurity [44]. This optimization algorithm provides a systematic and resource-efficient way to discover suitable mobile phases for radiopharmaceuticals and appears to have high versatility for a broad range of radiopharmaceuticals [44], suggesting that it will be possible to develop high-resolution TLC-based methods to purify other radiotracers. Despite the presence of various organic solvents in the TLC mobile phases, GC-MS analysis revealed the amounts to be minimal and far below permitted amounts (Table 3). The low values are likely due to (i) the low initial volume of mobile phase “contained” within the silica in the region of the product band, (ii) the application of heat (90 °C for 30 s) to dry the TLC plate after separation, and (iii) the use of vacuum during the sorbent collection step that may further assist in the removal of any residual solvents.

An additional requirement for more widespread use would be to increase the degree of automation to simplify the process and reduce radiation exposure, especially for producing clinical scale or multi-patient batches. Simplifications could be made in the process to reduce exposure, e.g., connecting the SPE tube to the sterilizing filter at the start of the experiment and pulling the vacuum through the sterile vent filter both for collecting the scraped silica into the SPE tube, as well pulling the extraction buffer through the silica. Further automation of each of the processes (sample deposition, TLC separation, and extraction of product) are also needed. While commercially-available systems exist for automated sample deposition in spots, lines, or other patterns (e.g., CAMAG automatic TLC sampler 4 [50]), transfer of the crude radiopharmaceutical to the device, operation time, and system footprint are concerns for use in radiochemistry applications. A more practical approach may be to simply use TLC plates with concentrating zones, which would allow the sample to be dripped at a controlled flow rate onto a single location onto a heated TLC plate rather than the more complicated process of depositing the sample in a streak pattern. The resolution obtained for [^18^F]PBR-06 samples spotted onto concentrating-zone HPTLC plates was nearly as good as for samples streaked onto normal analytical plates and could perhaps be further optimized by comparing different types of concentrating-zone plates. Concentrating zone plates are also likely to reduce the potential dispersion effects if the streak pattern is not perfectly straight. The need for manual handling in the development process can likely be eliminated by integrating the above sample deposition approaches with commercial or custom horizontal TLC setups [51,52,53,54]. Commercially available online extraction systems also exist for the collection of identified product bands directly from TLC plates without the need for scraping (e.g., CAMAG TLC-MS Interface 2 [55], Advion Plate Express [56]), using methods such as liquid extraction. However, the manual steps for installation and alignment of TLC plates, system size, and limitations on the band geometry (that will prevent complete collection of the product species) may not be well matched to preparative applications in the radiopharmaceutical field. A more practical approach to automation may be to develop a custom apparatus with adjustable or movable flow cell placed across the product band to extract the species of interest [57,58].

Another strategy for automation may be to leverage the PRISMA procedure to develop mobile phase systems that could provide high separation resolution using other chromatography methods (e.g., silica flash chromatography), which may be easier to automate, or perhaps miniaturize, using microfluidic-based systems with integrated purification media [32,59]. However, it is not clear if the resolution achieved in the column format would match that achieved in the planar TLC format or whether a similar fast operation speed and high recovery efficiencies would be observed. Furthermore, the use of highly-UV-absorbing organic solvents could limit the ability to monitor non-radioactive impurities and obtain a pure product, and the collected radiopharmaceutical would require extensive reformulation to remove relatively large amounts of solvents, making the process more time-consuming and complicated compared to the TLC-based approach.

## 4. Materials and Methods

### 4.1. Reagents and Materials

All reagents and solvents were obtained from commercial suppliers and used without further purification. 2,3-dimethyl-2-butanol (thexyl alcohol; anhydrous, 98%), acetic acid (AcOH; glacial, >99.9%), acetone (suitable for HPLC, >99.9%), acetonitrile (MeCN, anhydrous, 99.8%), ammonium formate (NH_4_HCO_2_, 97%), chloroform (>99.5%, contains 100–200 ppm amylenes as a stabilizer), dichloromethane (DCM; anhydrous, >99.8% contains 40–150 ppm amylene as a stabilizer), diethyl ether (Et_2_O; >99.9% inhibitor free), ethyl acetate (EtOAc; anhydrous, 99.8%), ethyl alcohol (EtOH; 200 proof, anhydrous, >99.5%), methyl alcohol (MeOH; anhydrous, 99.8%), n-hexanes (98%), Polypropylene SPE tube with PE frits (1 mL, 20 um porosity), Silica with concentrating zone (Silica 60 with diatomaceous earth zone) HPTLC plates, tetrahydrofuran (THF; anhydrous, >99.9% inhibitor free), water (H_2_O; suitable for ion chromatography) and Whatman Anotop 10 syringe filters (sterile, 0.2 um) were purchased from Sigma-Aldrich (St. Louis, MO, USA). (S)-2,3-dimethoxy-5-[3-[[(4-methylphenyl)-sulfonyl]oxy]-propyl]-N-[[1-(2-propenyl)-2-pyrrolidinyl]methyl]-benzamide ([^18^F]Fallypride precursor, >95%), 5-(3-fluoropropyl)-2,3-dimethoxy-N-(((2S)-1-(2-propenyl)-2-pyrrolidinyl)methyl)benzamide (Fallypride reference standard, >95%), 2-((2,5-dimethoxybenzyl)(2-phenoxyphenyl)amino)-2-oxoethyl 4-methylbenzenesulfonate ([^18^F]PBR-06 precursor, >95%), 2-fluoro-N-(2-methoxy-5-methoxybenzyl)-N-(2-phenoxyphenyl)acetamide (PBR-06 reference standard, >95%), and tetrabutylammonium bicarbonate (TBAHCO_3_; 75 mM in ethanol) were purchased from ABX Advanced Biochemical Compounds (Radeberg, Germany).

Silica gel 60 F_254_ sheets (aluminum backing, 5 cm × 20 cm) were purchased from Merck KGaA (Darmstadt, Germany). Glass microscope slides (76.2 mm × 50.8 mm, 1 mm thick) were obtained from C&A Scientific (Manassas, VA, USA). Saline (0.9% sodium chloride injection, USP) was obtained from Hospira Inc. (Lake Forest, IL, USA). Sodium phosphate dibasic (Na_2_HPO_4_-7H_2_O) and sodium phosphate monobasic (NaH_2_PO_4_-H_2_O) were purchased from Fisher Scientific (Thermo Fisher Scientific, Waltham, MA, USA).

No-carrier-added [^18^F]fluoride was produced by the (p, n) reaction of [^18^O]H_2_O (98% isotopic purity, Huayi Isotopes Co., Changshu, China) in a RDS-111 cyclotron (Siemens, Knoxville, TN, USA) at 11 MeV, using a 1.2-mL silver target with Havar foil.

### 4.2. Preparation of Radiopharmaceuticals and Reference Standards

[^18^F]PBR-06 and [^18^F]Fallypride were prepared using droplet radiochemistry methods on Teflon-coated silicon surface tension trap chips [26]. Detailed protocols for preparing these radiotracers have been previously reported [27]. Stock solutions of reference standards were prepared at 20 mM concentrations: 5 mg of Fallypride was added to 685 µL of MeOH, and 5 mg of PBR-06 was added to 632 µL of MeOH.

### 4.3. Preparation of TLC Plates

TLC plates were cut (W × H, 3 × 6 cm), then marked with horizontal pencil lines at 1 cm (origin line) and 5 cm (development line) from the bottom edge.

To eliminate impurities in the TLC plate that can contaminate the radiopharmaceutical, plates were pre-cleaned with solvent, as previously described [60]. Briefly, TLC plates were submerged to the origin line in a mixture of 2:1 EtOAc: MeOH (*v*/*v*), allowed to develop for 20 min, and then heated for 1 min (at a 120 °C setting) using a heat gun (Furno 500, Wagner).

### 4.4. Sample Spotting and Separation

60 µL of the relevant crude radiopharmaceutical sample was applied to the plate by various methods (e.g., sequential spotting or streaking) by a micro-pipette. Spotting on analytical scale TLC plates was performed by adding 1 µL of the sample and heating with a heat gun at 90 °C (~2 s). Spotting of samples on HPTLC plates occurred with the addition of 10 µL of sample to the concentrating zone, followed by drying at 90 °C (~5 s). Streaking of samples on analytical scale TLC plates were performed by deposition of 20 µL of sample in a thin streak covering ~30 mm, followed by heating at 90 °C (~5 s).

Plates were then developed in the mobile phase up to the development line. The mobile phases for [^18^F]PBR-06 and [^18^F]Fallypride were 29.8:26.9:20.4:22.85:0.05 (*v*/*v*) Et_2_O:DCM:CHCl_3_:n-hexanes:AcOH and 31.3:24.5:34.3:10.0 (*v*/*v*) THF:acetone:n-hexanes:TEA, respectively. After development, the plates were dried by a heat gun for 30 s at 90 °C.

### 4.5. Readout and Analysis of TLC Plates

The developed TLC plate was covered with a glass plate and visualized, as previously reported [35], to obtain a Cerenkov luminescence image (CLI) (1 min exposure), followed by a UV image (7 ms exposure).

Images were analyzed to determine chromatographic resolution using a custom MATLAB program (Mathworks, Natick, MA, USA) with a graphical user interface (GUI), as previously described [44]. Briefly, the user is guided by the program to create chromatograms from the CLI and UV images, from which peak positions, widths, and resolution are calculated [44]. In the analysis, the lines drawn (for origin and solvent front) are omitted from the selected lanes, since the pencil markings show up as false peaks in the UV chromatogram. The TLC chromatograms were plotted by exporting the data from the Matlab program and processing using OriginPro (OriginLab, Northampton, MA, USA).

### 4.6. TLC Purification of Radiopharmaceuticals

#### 4.6.1. Collection of Sorbent from TLC

When performing purification, the CLI and UV images of the TLC plate were used to identify the location of the product band and nearest impurity bands. During the preparation of the TLC plate, a pencil was used to outline the expected position and size of the radiopharmaceutical band (as determined from averaging images of multiple separations from crude batches of the same radiopharmaceutical and identifying the midpoint between the radiopharmaceutical band and its nearest impurities). To scrape the sorbent from the plate, the opening of a piece of plastic tubing cut at a ~45° angle (polyurethane, 1/4″ ID, IDEX) was used. The tubing was connected to the inlet of an empty SPE tube (polypropylene, 1 mL, Sigma Aldrich, St. Louis, MO, USA) that was fitted at the output end with a 10 µm frit (polyethylene, Sigma Aldrich), and the output end was further connected to vacuum. While the desired region was scraped in a series of horizontal lines (raster motion), the sorbent was collected into the SPE tube. The visualization step could be omitted through the pre-calibration step of determining the margins of radiopharmaceutical collection.

#### 4.6.2. Extraction of the Radiopharmaceutical from Sorbent

Before extraction, the sterile product vial was fitted with 2 sterile filters (Anotop, 0.2 µm), one prewetted with saline and then connected to the output of the SPE tube and one left dry (vent). The radiopharmaceutical was then eluted from the collected sorbent with biocompatible solvents (1 mL saline, or 100 µL EtOH followed by 900 µL saline) by applying vacuum to the vent filter of the product vial and by moving the tubing ‘scraper’ into an Eppendorf tube filled with the desired extraction solvent. No separate re-formulation of the collected purified product was required.

### 4.7. HPLC Analyses

Radio-HPLC was used to analyze crude radiopharmaceuticals and to perform tests for radiochemical and chemical purity and radiochemical identity of TLC-purified batches of radiopharmaceuticals. The radio-HPLC system setup comprised a Smartline HPLC system (Knauer, Berlin, Germany) equipped with a degasser (Model 5050), pump (Model 1000), UV detector (254 nm; Eckert & Ziegler, Berlin, Germany), gamma-radiation detector (BFC-4100, Bioscan, Inc., Poway, CA, USA), and counter (BFC-1000; Bioscan, Inc., Poway, CA, USA). A C18 Gemini column was used for separations (250 × 4.6 mm, 5 µm, Phenomenex, Torrance, CA, USA). [^18^F]PBR-06 samples were separated with a mobile phase of 60:40 (*v*/*v*) MeCN:20 mM sodium phosphate buffer (pH = 5.8) at a flow rate of 1.5 mL/min resulting in a retention time for [^18^F]PBR-06 of 6.5 min. [^18^F]Fallypride samples were separated with a mobile phase of 60% MeCN in 25 mM NH_4_HCO_2_ with 1% TEA (*v*/*v*) at a flow rate of 1.5 mL/min resulting in a retention time for [^18^F]Fallypride of 5.8 min.

### 4.8. Quality Control Testing

Quality control (QC) tests were performed on 3 consecutive batches of [^18^F]PBR-06 produced via a droplet microreactor and purified with the TLC approach described here. Testing focused primarily on color and clarity, radiochemical and chemical purity, molar activity, and residual solvent content to highlight the performance of this novel purification method. A full summary of tests and results can be found in the Appendix A.

### 4.9. ICP-MS Analysis for Silicon Content

To estimate silica content in the final formulation, the amount of silicon was determined in a series of replicate samples, in which the spotting, separation, silica collection and extraction steps (using 1 mL saline) were performed starting with blank TLC plates. Silicon determination was performed via inductively-coupled plasma mass spectrometry (ICP-MS) using a NexION 2000 (Perkin Elmer, Hong Kong, China). For each sample, an area (2.0 × 1.5 cm, W × H) was scraped from a cleaned TLC plate into an SPE tube, and 1 mL of saline was flowed through the silica and a sterile filter and collected into an Eppendorf tube for analysis. Each sample was transferred to a clean Teflon vessel for acid digestion in concentrated HNO_3_ (65–70%, Trace Metal Grade, Fisher Scientific) with a supplement of H_2_O_2_ (30%, Certified ACS, Fisher Scientific) at 200 °C for 50 min in a microwave digestion system (Titan MPS, Perkin Elmer). Once the sample was cooled to room temperature, it was subsequently diluted to make a final volume of 10 mL by adding filtered DI H_2_O for analysis. The calibration curve was established using a standard solution, while the dwell time was 50 ms with thirty sweeps and three replicates with background correction. The detection limit using this procedure was 0.82 ng/mL.

## 5. Conclusions

In this feasibility study, high-resolution radio-TLC was leveraged as a means to perform rapid purification of two clinically-relevant radiopharmaceuticals ([^18^F]PBR-06 and [^18^F]Fallypride) produced via droplet radiochemistry methods. Due to the high chemical and radiochemical purity and the high efficiency of product collection and formulation achieved, it is conceivable that the TLC purification method could serve as a versatile approach for the purification of microscale-produced radiopharmaceuticals. The combination of droplet radiosynthesis with TLC-based purification/formulation for the production of [^18^F]PBR-06 led to high molar activities (⪎300 GBq/µmol), comparing favorably to the literature reports (37–222 GBq/μmol [61,62]).

Even with the higher mass loading and volume of the crude radiopharmaceutical (60 µL) compared to typical samples (0.5–1 µL), high separation resolution of the radiopharmaceutical product from radioactive and non-radioactivity impurities was achieved on the TLC plates, as visualized via CLI and UV imaging. The product collection (via sorbent collection from the plate followed by extraction) was nearly quantitative. Notably, by using injectable buffers (saline or EtOH diluted to <10% *v*/*v* in saline), the need for subsequent re-formulation is eliminated. Consequently, radio-TLC purification (and formulation) could be completed in under 10 min. Furthermore, due to the low cost of TLC plates, one can consider the purification and formulation system to be disposable (in stark contrast to HPLC-based systems), further simplifying microscale radiosynthesis instruments and eliminating the need for developing and validating cleaning protocols

As a proof-of-concept, several batches of [^18^F]PBR-06 and [^18^F]Fallypride were produced and purified at scales sufficient for clinical imaging. Critical QC tests were performed on multiple batches (e.g., color and clarity, chemical and radiochemical purity, molar activity, and residual solvents) and suggested the potential suitability for clinical production of the TLC purification method.

## Figures and Tables

**Figure 1 molecules-27-08178-f001:**
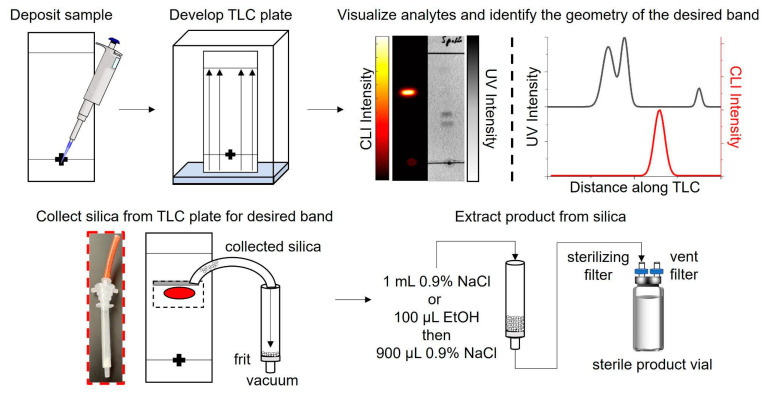
Procedure for the purification of microscale-synthesized radiopharmaceuticals using TLC.

**Figure 2 molecules-27-08178-f002:**
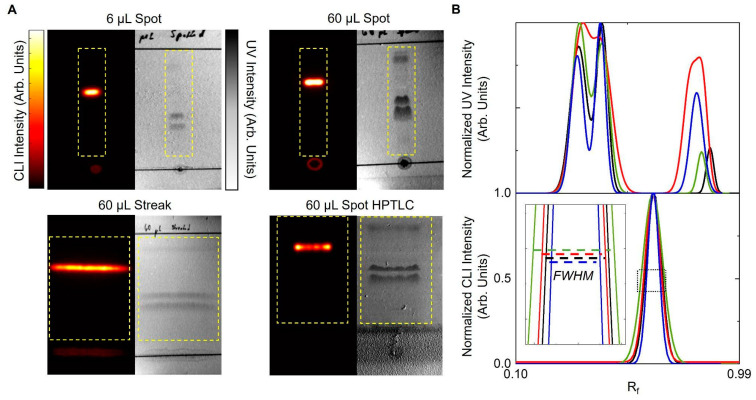
Effect of sample deposition parameters on separating [^18^F]PBR-06 samples. (**A**) Images (left: CLI; right: UV) of crude [^18^F]PBR-06 deposited on TLC plates using different volumes and application methods. Yellow lines denote the area of the image used to compute the line profiles shown in panel B, excluding the origin and solvent front lines with a strong signal in the UV images. (**B**) TLC chromatograms generated from the CLI and UV images. Legend: black—6 μL spot, red—60 μL spot, green—60 μL streak, and blue—60 μL spot (HPTLC plate). The inset shows a magnified view of the dashed region to highlight the full-width half maximum (FWHM).

**Figure 3 molecules-27-08178-f003:**
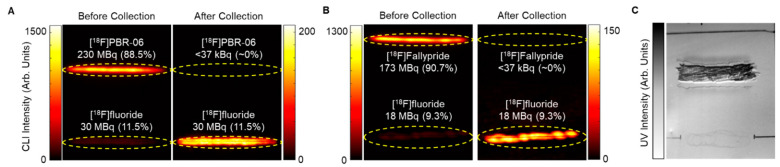
CLI images of TLC plates show the effectiveness of the stationary-phase removal step during TLC-based purification. (**A**) Images of analytical TLC plate streaked with crude [^18^F]PBR-06 before and after collection. (**B**) Images of analytical TLC plate streaked with crude [^18^F]Fallypride before and after collection. Yellow bands denote ROIs used in quantifying the proportion of different radiochemical species. (**C**) UV image of an analytical TLC plate after stationary phase removal for recovery of [^18^F]PBR-06.

**Figure 4 molecules-27-08178-f004:**
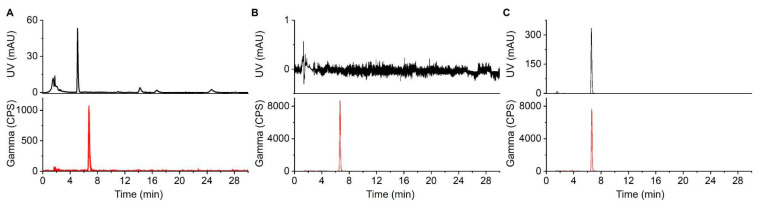
HPLC chromatograms of [^18^F]PBR-06. (**A**) Crude reaction mixture. (**B**) TLC-purified (and formulated) product. (**C**) Co-injection of TLC-purified product with the [^19^F]PBR-06 reference standard.

**Table 1 molecules-27-08178-t001:** Effect of sample deposition parameters on chromatographic resolution between [^18^F]PBR-06 and the nearest impurity.

Sample Volume (μL)	Deposition Method	TLC Plate	Resolution
6	Spot	Analytical	2.2
60	Spot	Analytical	1.0
60	Streak	Analytical	2.0
60	Spot	HPTLC (with concentrating zone)	1.7

**Table 2 molecules-27-08178-t002:** Performance of microscale droplet radiosyntheses coupled with the TLC-based purification and formulation. In the extraction step, Method 1 uses 1.0 mL of saline alone, and Method 2 uses 100 µL EtOH, followed by 900 µL saline. The overall collection and extraction efficiency is calculated by multiplying the silica collection efficiency by the extraction efficiency for individual runs and then averaging across replicates. The overall RCY is calculated by multiplying the crude RCY of the droplet synthesis by the silica collection efficiency and the extraction efficiency for individual runs and then averaging across replicates.

Radiotracer	Activity Level (MBq)	Crude RCY of Droplet Synthesis (%) (*n* = 8)	Silica Collection Efficiency (%) (*n* = 8)	Extraction Efficiency (%)		Overall Collection and Extraction Efficiency (%)		Overall RCY (%)	
				Method 1 (*n* = 4)	Method 2 (*n* = 4)	Method 1 (*n* = 4)	Method 2 (*n* = 4)	Method 1 (*n* = 4)	Method 2 (*n* = 4)
[^18^F]PBR-06	11	94.4 ± 1.2	98.7 ± 1.3	96.4 ± 3.4	97.9 ± 1.6	95.4 ± 4.6	96.3 ± 1.7	89.6 ± 3.9	91.3 ± 1.9
	1110–1480	91.9 ± 1.8	98.1 ± 1.1	95.6 ± 2.9	98.2 ± 0.3	94.2 ± 2.6	95.9 ± 0.9	86.7 ± 3.7	87.9 ± 1.8
[^18^F]Fallypride	7.5	96.5 ± 1.6	97.5 ± 1.6	95.4 ± 1.1	98.4 ± 0.3	92.6 ± 2.6	96.2 ± 1.3	89.4 ± 3.7	92.9 ± 2.6
	740–1480	93.2 ± 2.5	97.5 ± 1.2	97.1 ± 1.0	97.8 ± 1.4	94.5 ± 1.9	95.6 ± 2.8	88.1 ± 3.8	89.2 ± 4.7

**Table 3 molecules-27-08178-t003:** Performance and quality control testing results for three consecutive batches of [^18^F]PBR-06.

Test	Criteria	Batch 1	Batch 2	Batch 3
Radioactivity	-	821 MBq [22.2 mCi]	744 MBq [20.1 mCi]	829 MBq [22.4 mCi]
Molar Activity	-	342 GBq/µmol	315 GBq/μmol	327 GBq/μmol
Appearance	Clear, colorless, and particulate-free	✓	✓	✓
Radiochemical Identity	Retention time ratio of radio peak vs. reference standard (0.90–1.10)	1.01	1.01	1.01
Residual TBAHCO_3_	<104 mg/L	<45 mg/L	<45 mg/L	<45 mg/L
Residual Solvents	MeCN < 410 ppm	<1	<1	<1
	MeOH < 3000 ppm	24	21	24
	Hexanes < 290 ppm	6	2	5
	CHCl3 < 60 ppm	<1	<1	<1
	Et2O < 5000 ppm	104	47	102
	EtOAc < 5000 ppm	21	10	20
	AcOH < 5000 ppm	7	5	7
	Thexyl alcohol < 5000 ppm	<1	<1	<1
Radiochemical Purity	>95%	>99%	>99%	>99%
Radionuclide Identity (half-life)	105–115 min	110.4	111.7	113.8
pH	4.5–7.5	5.5	5.5	5.5
Shelf life	Pass appearance, pH, and radiochemical purity after 120 min	✓	✓	✓

## Data Availability

The data presented in this study are available on request from the corresponding author. The data are not publicly available due to the length of the datasets.

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
