# Peer review of "Rapid Purification and Formulation of Radiopharmaceuticals via Thin-Layer Chromatography"

_molecules, 2022, doi:10.3390/molecules27238178_

Round 1

Reviewer 1 Report

The paper by Laferriere-Holloway et al. is a well-structured and interesting report on using a seldom used purification method for radiopharmaceuticals. Overall the paper could make a significant impact in the field, if the challenges or automation and/or robustness are solved.

Find below few comments that could improve the paper:

l13: not always the impurities are toxic, amend this statement and the other occurrences as needed

l58-59: in some instances, a bio-compatible mobile phase can be used, although pressure rating might be an issue sometimes

l116: are there any issues with heating the spots at 120C? Said in a different way, can you assess if the application process degrades the product by comparing the RCY of the solution before and after spotting it?

l159: I imagine the process of scraping is a pretty manual one requiring operator skills. Also, in the nuclear industry, the use of radioactive solids (dry powders) is not seen well and needs extensive safety measures to reduce likelihood of operator inhalation. Can the authors comment on safety measures to be taken with this proposed approach; this aspect may become crucial when handling higher activities, but it could be useful to point out the key hazards and measures

Fig 3: it could help also to have an optical picture of the "scraped-off" TLC plate

l232: I am curious to know if the authors can comment on how to automate the TLC purification; in particular, if there are already (probably oversized) commercial solutions for it.

l267 and following: the way the "Discussion" section starts look more a Conclusion. Maybe better to reduce the claims, and focus on discussing their validity.

l289: the application of PRISMA algorithm is indeed intriguing and possibly new in the radiopharmaceutical space. I wonder if such approach could be used to develop a low-pressure Si-based chromatographic separation of the products. After all, TLC generally provide a guide of which solvent mixtures to use for flash-chromatography on Si. This possibility will still need some minor additional steps (e.g. solvents evaporation), but would be much easier to automate. On the same point, maybe it would be interesting or educationally useful to show how the optimization algorithm has been used for this project, even maybe in the supp info, as this would be a crucial information if another scientist would like to adopt this approach.

l298: is it 90 or 120C?

l317: (cite) is not needed?

l425: I am curious to know what the rationale is in using the EtOH-saline sequence, instead of only saline, given that the sorbent is silica and not others (eg C18). Was there any knowledge that the products would stick to silica?

l437: was it a Gemini or a Kinetex used?

l451: it is not clear if the test for silicon content was performed on each of the tracers' batches, or on test conditions only

Author Response

We thank the reviewer for their comments regarding our work. Below we have included responses for each comment made:

  1. l13: not always the impurities are toxic, amend this statement and the other occurrences as needed

We appreciate the reviewer’s comment and have adjusted the text to reflect this.

  1. l58-59: in some instances, a bio-compatible mobile phase can be used, although pressure rating might be an issue sometimes

We agree with the reviewer’s comment, and have updated the Introduction to reflect this

  1. l116: are there any issues with heating the spots at 120C? Said in a different way, can you assess if the application process degrades the product by comparing the RCY of the solution before and after spotting it?

Based on a comment later raised, we have corrected the temperature (it should be 90 C). Further, we have added a sentence in Section 2.1 to reflect that there is no decomposition of the products observed when using heat to dry the samples on the TLC plates.

  1. l159: I imagine the process of scraping is a pretty manual one requiring operator skills. Also, in the nuclear industry, the use of radioactive solids (dry powders) is not seen well and needs extensive safety measures to reduce likelihood of operator inhalation. Can the authors comment on safety measures to be taken with this proposed approach; this aspect may become crucial when handling higher activities, but it could be useful to point out the key hazards and measures.

We appreciate the reviewer’s comment on this point and agree that the creation and handling of radioactive powders do present certain safety concerns, especially at higher activities and/or in cases of longer-lived isotopes. We have added a sentence in Section 2.2 clarifying that we used vacuum during the scraping process to reduce this concern. With the vacuum-assisted manual process, we observed very low activity losses, suggesting that, in the worst case, only a tiny fraction of the sorbent is lost. In the longer term, we think automation of band extraction using other methods not based on scraping, e.g., flowing an extractant across a localized part of the plate surface, will be preferable.

  1. Fig 3: it could help also to have an optical picture of the "scraped-off" TLC plate

We appreciate the reviewer’s comment and have added a brightfield image of a scraped plate to Figure 3.

  1. l232: I am curious to know if the authors can comment on how to automate the TLC purification; in particular, if there are already (probably oversized) commercial solutions for it.

We thank the reviewer for these comments. We have added to the discussion section describing the use of commercially available systems for sample deposition, development, and collection from manufacturers such as CAMAG. However, these systems are quite bulky and require additional specialized systems or user intervention to transfer the plate between instruments for different steps. These factors limit the practicality of existing commercial solutions in radiochemistry applications. We have elaborated on this in the discussion.

  1. l267 and following: the way the "Discussion" section starts look more a Conclusion. Maybe better to reduce the claims, and focus on discussing their validity.

We thank the reviewer for this comment and agree. We have re-ordered the discussion section and moved these points into the conclusions.

  1. l289: the application of PRISMA algorithm is indeed intriguing and possibly new in the radiopharmaceutical space. I wonder if such approach could be used to develop a low-pressure Si-based chromatographic separation of the products. After all, TLC generally provide a guide of which solvent mixtures to use for flash-chromatography on Si. This possibility will still need some minor additional steps (e.g. solvents evaporation), but would be much easier to automate. On the same point, maybe it would be interesting or educationally useful to show how the optimization algorithm has been used for this project, even maybe in the supp info, as this would be a crucial information if another scientist would like to adopt this approach.

We thank the reviewer for these comments.

Regarding the first point, we have added a brief section to our discussion to describe the possibility of using PRISMA to develop mobile phase systems for other chromatography methods as an alternative path toward automation. In general, we think it could be feasible, though there are questions about the performance (speed, resolution, etc.) and the need for additional steps, e.g. a separate formulation step to remove solvents, as pointed out.

Regarding the second point, we do believe the use of PRISMA in radiochemistry is novel. We have a separate manuscript under review, which describes in detail the method to develop the mobile phases used for separation of [18F]Fallypride and [18F]PBR-06. We intend to ask the editor to hold off on publication briefly so that we can cite that paper in this manuscript. (For now we have cited a conference abstract where we presented the use of the PRISMA method.)

  1. l298: is it 90 or 120C?

Thank you for catching this inconsistency. Indeed it should be 90C. (We had performed initial studies at 120C setting and later determined it could be performed at 90C.) We have corrected this throughout.

  1. l317: (cite) is not needed?

Thank you for pointing out this typo, leftover from the editing process. We removed the text “(cite)”.

  1. l425: I am curious to know what the rationale is in using the EtOH-saline sequence, instead of only saline, given that the sorbent is silica and not others (eg C18). Was there any knowledge that the products would stick to silica?

Based on our experience that some radiopharmaceuticals, when formulated via evaporation, require additives such as EtOH or other to enhance solubility during reconstitution into saline, we wanted to compare whether such an approach (if needed for other radiopharmaceuticals), would create problems here.

  1. l437: was it a Gemini or a Kinetex used?

We have corrected this error to reflect that a Gemini column was used

  1. l451: it is not clear if the test for silicon content was performed on each of the tracers' batches, or on test conditions only

Due to the use of centralized campus facilities where we can’t control the timing of sample analysis, we wanted to avoid long-term storage of samples that could potentially change the silicon content, e.g. due to effects of freezing/thawing or leaching from containers. Instead we used samples that could be prepared on very short notice when the core facility alerted us they were ready for our samples, and thus used test conditions. We performed all process steps for these sample batches, including spotting of crude solvent, TLC separation, silica scraping, and extraction using 1 mL saline (i.e. identical to conditions used in radioactive experiments). We have updated section 4.9 to clarify.

Reviewer 2 Report

Review: Rapid purification and formulation of radiopharmaceuticals 2 via thin-layer chromatography

This paper reports the development of using TLC as purification method for radiopharmaceuticals produced using microfluidics.  

The paper is very well written. The main questions this reviewer had going in was:

1.       Organic impurities in the formulation.

2.       Silica impurities in the formulation.

3.       Radiation safety and automation of the process.

The authors answers point one and two with data and gives a thorough discussion about point three.

Author Response

We appreciate the kind words from the reviewer regarding this work.